# Chemical Constituents and Molecular Mechanism of the Yellow Phenotype of Yellow Mushroom (*Floccularia luteovirens*)

**DOI:** 10.3390/jof8030314

**Published:** 2022-03-18

**Authors:** Xiaolong Gan, Xuemei Bao, Baolong Liu, Yun Li, Dong Cao, Hg Zhang, Yuan Zong

**Affiliations:** 1Qinghai Province Key Laboratory of Crop Molecular Breeding, Xining 810008, China; ganxiaolong@nwipb.cas.cn (X.G.); baoxuemei@nwipb.cas.cn (X.B.); bliu@nwipb.cas.cn (B.L.); liyun@nwipb.cas.cn (Y.L.); caodong@nwipb.cas.cn (D.C.); 2Key Laboratory of Plant Resources Conservation and Sustainable Utilization, South China Botanical Garden, Chinese Academy of Sciences, Guangzhou 510650, China; 3University of Chinese Academy of Sciences, Beijing 100049, China; 4School of Education, Qinghai Normal University, Xining 810008, China

**Keywords:** *Floccularia luteovirens*, riboflavin, transcriptome, transporter, *FlMCH5*, transgenic

## Abstract

(1) Background: Yellow mushroom (*Floccularia luteovirens*) is a natural resource that is highly nutritional, has a high economic value, and is found in Northwest China. Despite its value, the chemical and molecular mechanisms of yellow phenotype formation are still unclear. (2) Methods: This study uses the combined analysis of transcriptome and metabolome to explain the molecular mechanism of the formation of yellow mushroom. Subcellular localization and transgene overexpression techniques were used to verify the function of the candidate gene. (3) Results: 112 compounds had a higher expression in yellow mushroom; riboflavin was the ninth most-expressed compound. HPLC showed that a key target peak at 23.128 min under visible light at 444 nm was Vb2. All proteins exhibited the closest relationship with *Agaricus bisporus var. bisporus* H97. One riboflavin transporter, CL911.Contig3_All (FlMCH5), was highly expressed in yellow mushrooms with a different value (log_2_ fold change) of −12.98, whereas it was not detected in white mushrooms. FlMCH5 was homologous to the riboflavin transporter MCH5 or MFS transporter in other strains, and the FlMCH5-GFP fusion protein was mainly located in the cell membrane. Overexpression of FlMCH5 in tobacco increased the content of riboflavin in three transgenic plants to 26 μg/g, 26.52 μg/g, and 36.94 μg/g, respectively. (4) Conclusions: In this study, it is clear that riboflavin is the main coloring compound of yellow mushrooms, and *FlMCH5* is the key transport regulatory gene that produces the yellow phenotype.

## 1. Introduction

*Floccularia luteovirens* (yellow mushroom), also known as *Armillaria luteovirens*, *Tricholoma luteovirens*, and golden mushrooms, belongs to the genus Tricholoma of phylum Basidiomycota and family *Agaricaceae*, which mainly grows in Qinghai, Sichuan, Tibet and other high-altitude areas [1,2]. Yellow mushroom is expensive due to its high nutrient contents, delicious taste, and scarcity. Altitude, season, environment, and other conditions limit the domestication and cultivation of yellow mushrooms (*Floccularia luteovirens*) [1]. Once the growth environment of yellow mushrooms is artificially damaged, it is difficult to repair [3]. At present, research on yellow mushroom is mainly focused on its nutritional characteristics, its effects on grassland vegetation and soil, its fermentation conditions, and the determination of trace elements and fatty acids [4,5,6,7]. It has been reported that the contents of eight kinds of essential amino acids, particularly lysine and arginine, are higher in yellow mushroom compared with those in *Pleurotus eryngii* and *Flammulina velutipes*, which are frequently consumed by humans [8,9]. In addition, yellow mushroom also contains 17 minerals, with K, P, and S being predominant, followed by Fe, Zn, and Cu. Se enrichment also increases the nutritional value of yellow mushroom [2,10]. As it is a food with high nutritional value, research on yellow mushroom is imminent, but the molecular mechanisms of the yellow phenotype are still unclear. So far, a number of studies have reported that carotenoids, flavonoids, and other compounds contribute to the yellow flower color or fruit color of angiosperms, but few studies focus on fungi [11,12]. In the absence of chloroplasts, the yellow phenotype of yellow mushroom may be quite different [13,14].

Riboflavin, also known as vitamin B2, is one of the water-soluble vitamins necessary for the human body, and its deficiency can lead to oral ulcers [15,16]. The human body itself cannot synthesize riboflavin and needs to obtain it from the outside world [17]. Riboflavin exists in various organs of the human body, promoting metabolism and normal vision [18]. Thus, riboflavin plays an important role in cell proliferation and growth [19]. Riboflavin is an isopropylamine derivative with a side chain of ribitol, mainly formed from the direct precursors GTP and ribulose 5-phosphate through a series of reactions catalyzed by riboflavin synthase [20]. GTP cyclohydrolase II catalyzes the ring-opening hydrolysis reaction of GTP imidazole to generate dopamine and adenosine 3′5′-monophosphate-regulated phosphor-protein (DARPP) [21]. Riboflavin synthase β-subunit catalyzes the conversion of DHBP and 5-amino-6-ribol amino-2,4 (1H, 3H)-pyrimidine Dione (ArP) to 6,7-dimethyl-8-ribitol-2,4-dioxatetrahydrobiopterin (DRL). Finally, DRL is directly converted to riboflavin [22]. Two molecules of DRL undergo a disproportionation reaction under the action of riboflavin synthase α subunit to produce one molecule of ArPP and one molecule of riboflavin [21,23].

Among the many membrane proteins, a large protein of transporters, known as the major facilitator superfamily (MFS), has been found to have 12 transmembrane helix domains [24]. MCH5 was homologous to mammalian monocarboxylate transporters, and the MFS structural domain it contains is the main reason for its transport function. Transport experiments in *S. cerevisiae* and *Schizosaccharomyces* pombe have revealed that Mch5p is a high-affinity transporter (Km = 17 μm) with an optimal pH of 7.5 and that the expression of MCH5 is regulated by the riboflavin content in cells [25]. This result indicates that *S. cerevisiae* possesses a mechanism for sensing riboflavin and avoiding riboflavin deficiency by increasing the expression of the plasma membrane transporter MCH5 [26]. Additionally, MFS transporters also occupy a very important position in plants, including plant P transporters, which mediate the absorption and transport of P by roots and maintain the steady state of P in plants [27,28,29].

In the present study, metabolomic analysis was used to understand the differences between the main compounds in yellow mushroom (*Floccularia luteovirens*) and white mushroom (*Agaricus campestri*s), as well as the major chemical constituents responsible for the yellow phenotype. At the same time, the major yellow compound was separated by HPLC. Second-generation transcriptome sequencing can quantify gene expression, mine transcripts, and provide reliable information to study molecular anabolic pathways in plants. Transcriptome-sequencing technology was employed to determine the major differences in the riboflavin anabolic pathway in the formation of the yellow phenotype of mushrooms and screen out the key gene that regulates the generation of the yellow and white differential phenotypes of *Floccularia luteovirens*.

## 2. Results

### 2.1. Analysis of Chemical Contents in Yellow Mushrooms

Overlapping display analysis of total ion chromatogram (TIC) for MS detection and analysis of different quality-control samples revealed sufficient reproducibility and reliability of the data for further investigation (Appendix A). A total of 573 compounds from 22 classes were detected (Appendix A), among which 227 compounds from 20 classes exhibited significantly different levels of accumulation in the yellow and white mushrooms (Appendix A). The contents of 112 compounds were higher in yellow mushroom (Figure 1a). In addition, one quinone and five steroids were detected in both yellow and white mushrooms, which did not exhibit differential accumulation (Table 1). The contents of 11 classes of compounds in yellow mushrooms were higher than those in white mushrooms. Among them, flavone, flavonol, isoflavone, carbohydrates, terpene, vitamins and their derivatives, and lipids were more than two-fold higher in yellow mushrooms, with flavone presenting more than eight-fold higher content compared with that in white mushrooms (Table 1). In addition, the contents of both flavonol and vitamins and their derivatives were significantly different between the two species of mushrooms. Three differentially accumulated flavonol compounds, dihydromyricetin and ayanin, were only found in the white mushrooms (Appendix A). A relatively high content of alcohol species was observed in white mushrooms, whereas little or no alcohol content was noted in yellow mushrooms with a log_2_ fold change between −3 and −17. In contrast, the contents of alkaloids and indole derivatives were higher in white mushrooms than those in yellow mushrooms.

Curcumin, a differentially expressed yellow-colored polyphenol, was only found in yellow mushrooms, with the log_2_ fold change value reaching 11.43, although its quantitative value was low. It was found that the content of lysoPC 16:0 in yellow mushrooms was the highest, followed by galactinol and azelaic acid, but they were not yellow compounds. When sorting and screening with the peak area as the estimated content value, it was found that riboflavin held ninth place in terms of content and was dominant in both mushroom species. The riboflavin content of yellow mushroom was high, with the log_2_ fold change value reaching 1.53.

### 2.2. Identification and Content Determination of Riboflavin

After the water extraction of 5 g of fresh yellow mushrooms, HPLC results showed a key target peak at 23.128 min under visible light at 444 nm. Subsequent HPLC detection revealed that the yellow water and riboflavin standard (Vb2) were the same, with a peak at 23.126 min (Figure 1b). Quantitative detection of riboflavin in white and yellow mushrooms using the national standard method showed that the white mushrooms extract appeared dark yellow after water extraction, whereas the yellow mushrooms changed from yellow to white after water extraction, with the extract appearing bright yellow. The white and yellow mushrooms showed consistent peaks at 444 nm; retention times of 7.935 and 7.934 min; peak areas of 10,262,820 and 16,221,615; and riboflavin contents of 0.89 and 1.35 mg/100 g, respectively (Figure 1c). The results of riboflavin content were consistent with the difference in the riboflavin content in metabolome sequencing.

### 2.3. Transcriptome Sequencing and Data Analysis

The total number of bases obtained for each sample after transcriptome sequencing was 8.31 Gb, which reduced to 7.54 Gb after filtering (Appendix A). The clean reads were further assembled into 48,997 unigenes, and after removing redundant genes, 43,750 genes were obtained. A BlastX search attained a total of 43,750 predicted proteins (Appendix A). Protein databases such as Nr, Nt, Swissprot, KEGG, KOG, Interpro, GO, and Intersection were used to annotate 35,796, 13,529, 21,680, 24,808, 21,199, 24,460, 6899, and 2348 genes, respectively (Figure 2a). Homology comparison revealed that all the proteins in the mushroom samples exhibited the closest relationship with Agaricus bisporus var. bisporus H97 (20.32%), followed by *Agaricus bisporus* var. *burnettii* JB137-S8 (17.25%) (Appendix A).

A total of 3863 unigenes were downregulated, and 2309 unigenes were upregulated in white mushrooms (Figure 2b). Through KEGG annotation, all the differentially expressed genes (DEGs) were annotated into 21 types of pathways and could be divided into 5 categories (Appendix A). Subsequently, 11 structural genes involved in riboflavin biosynthesis and metabolism and 60 homologous genes were selected. Among them, five structural genes were not differentially expressed, or their expression levels were low in yellow mushrooms (Appendix A). The rest of the six structural genes were upregulated at different levels in yellow mushrooms, of which the transcripts of three structural genes, rib3, rib4, and FLAD1, were only found in yellow mushrooms but not in white mushrooms (Figure 2c). Furthermore, the transcript of the homologous gene of riboflavin transporter CL911.Contig3_All (FlMCH5) was highly expressed in yellow mushrooms, with a log_2_ fold change value of −12.98, whereas it was not detected in white mushrooms (Appendix A). Therefore, FlMCH5 was screened as the key regulatory gene of the yellow phenotype for function-prediction analysis.

### 2.4. Conjoint Analysis of Transcriptome and Metabolome

The riboflavin cell transport pathway showed that riboflavin was upregulated by 1.54-fold in yellow mushrooms, whereas the downstream product FAD was downregulated by 1.25-fold, and the resultant glutamic acid was upregulated 1.01-fold (Figure 2c). To further evaluate the effects of transcriptome change on the metabolome, the DEGs and differentially accumulated compounds were investigated using the KEGG database. A total of 66 pathways were determined to possess both DEGs and differentially accumulated compounds, and the *p* value was used as the limit to determine the enrichment of genes or compounds. DEGs were mainly enriched in seven metabolic pathways, namely, Aminoacyl-tRNA Biosynthesis, Yeast Cell Cycle, Cysteine and Methionine Metabolism, Longevity-Regulating Pathway—Multiple Species, Lysine Degradation, Propanoate Metabolism, and Riboflavin Metabolism. Differentially accumulated compounds were mainly enriched in two metabolic pathways, namely, Glycerophospholipid Metabolism and Purine Metabolism (Appendix A). The CL911.Contig3_All gene was involved in Glycine, Serine, and Threonine Metabolism; Beta-Alanine Metabolism; Phenylalanine Metabolism; and Tyrosine Metabolism pathways (Appendix A).

### 2.5. Molecular Characteristics of FlMCH5

Phylogenetic tree analysis revealed that the CL911.Contig3_All gene was homologous to the riboflavin transporter MCH5 or MFS transporter in other species (Figure 3a). Bioinformatics analysis showed that the 1–301 amino acid sequence of the CL911.Contig3_All gene belongs to the MFS, with structural domains similar to CfMFS (*Crassisporium funariophilum*) and HmMCH5 (*Hypsizygus marmoreus*) (Figure 3b). Among the CL911.Contig3_All gene protein sequences, the widest coverage was exhibited by the 7bp3.1.A protein, which belonged to the MFS and was a homodimer. These results confirmed that the CL911.Contig3_All gene was very likely to participate in importing riboflavin in yellow mushrooms.

The FlMCH5-GFP fusion cistron was constructed from cauliflower mosaic virus 35S promoter (pK7WGF-2) and was transiently expressed in tobacco cells. The results showed that the FlMCH5-GFP fusion protein was mainly located in the cell membrane, while the control GFP protein was distributed throughout the cell (Figure 3c).

### 2.6. Overexpression of FlMCH5 Induces Riboflavin in Tobacco

The PC2300: FlMCH5 plasmid was transferred into Agrobacterium strain GV3101 using the freeze–thaw method. The Agrobacterium-mediated leaf disk transformation method was used to obtain transgenic tobacco. For further experiments, the T3 family lines carrying the objective gene without separation were used. The color of the transgenic tobacco leaf extract is darker than the wild type. Using Vb2 standard products to calculate the content of riboflavin in tobacco, we obtained totals of 4.25 μg/g (WT), 26 μg/g (FlMCH5-1), 26.52 μg/g (FlMCH5-2), and 36.94 μg/g (FlMCH5-3) (Figure 3d). The relative riboflavin concentration of the transgenic lines was much higher than that of the wild type. These results showed that FlMCH5 could activate riboflavin biosynthesis by acting as an MFS factor in tobacco.

## 3. Discussion

Currently, research on yellow mushrooms is mainly focused on the determination and analysis of trace elements, whereas epigenetic investigation is still scarce. In the present study, chemical analysis suggested that the yellow phenotype of yellow mushrooms is the result of riboflavin accumulation. Stepwise MIM-EPI and RNA-Seq were confirmed to be effective strategies for performing comprehensive metabolomic and transcriptomic analyses of yellow and white mushrooms, respectively. A total of 573 compounds from 22 classes were detected, and 43,750 unigenes were assembled.

### 3.1. Metabolome Clarify the Main Compounds in Mushrooms

Principal Component Analysis (PCA) showed significant differences between the yellow and white mushrooms (Appendix A). OPLS-DA revealed that the data point was farthest from the origin, indicating that the greater the contribution of this point to separation between the groups, the better the correlation. The vast majority of the distances shown in the figure are red, upregulated compounds, signifying that the yellow phenotype of yellow mushrooms is mainly caused by the upregulated compounds (Appendix A). The reason for the significant difference in the flavone content between the two varieties of mushrooms was mainly the presence of sakuranetin, C-hexosyl-luteolin O-hexosyl-O-pentoside, tricin 7-O-hexosyl-O-hexoside, and tricin 5-O-hexosyl-O-hexoside in yellow mushrooms, which were almost undetectable in white mushrooms. The metabolomic analysis also found some compounds in yellow mushrooms, such as vanillic acid, 6-hydroxy-4-methylcoumarin, skimmine, curdione, α-ionone, phytocassane D, and eudesmic acid, that were not detected in white mushrooms (Appendix A). These compounds have high medicinal values, including antibacterial, antitumor, anti-inflammatory, and antioxidant properties and enhance the health benefits of yellow mushrooms [30,31,32,33,34].

The highest order of compound content is mainly lipids, organic acids and derivatives, and vitamins and derivatives, but among these compounds, only riboflavin is yellow, and the other compound monomers are white powder. Therefore, the high riboflavin content is likely to be the chromogenic substance of the yellow mushroom phenotype. The quantification results of riboflavin in Cantharellus were consistent with those of the metabolomic analysis, revealing that the riboflavin content was higher in the yellow mushrooms. Interestingly, riboflavin could also be extracted from white mushrooms. The extract appeared yellow in color, further suggesting that riboflavin in white mushrooms is likely to be inside the cell and not transported outside the cell, resulting in sufficient riboflavin content but not the yellow phenotype in white mushrooms.

### 3.2. Riboflavin Is the Main Chemical Basis of the Yellow Phenotype

The wavelength of visible light is in the range of 400–760 nm. In nature, most of the major yellow compounds in yellow flowers, fruits, or tubers are carotenoids [35]. If an object absorbs violet light at a wavelength of 400–435 nm, then it will appear yellow-green in color [36]. The absorption ratio of chlorophyll and carotenoid is the highest at 400–520 nm (blue light), which has the maximum impact on photosynthesis [37]. Therefore, 444 nm was selected for scanning the compound that causes the yellow phenotype of yellow mushroom. A white phenotype was observed after soaking the yellow mushrooms in water, and the extract obtained was yellow in color and translucent, indicating that the compounds that cause the yellow phenotype were easily soluble in water (Figure 1c). HPLC analysis of the yellow mushroom extract at 444 nm revealed two main absorption peaks at 2.8 and 23.128 min, with the first 5 min absorption peak considered as the solvent peak. Therefore, it was speculated that the yellow phenotype might have been caused by the compound that appeared at 23.126 min. Although there were other sensitive peaks at 5–10 min intervals, the sensitive voltage value was lower. Subsequently, the compound that appeared at 23.126 min was eluted using acetonitrile, concentrated, and dried to obtain a yellow powder. Previously, 1H-NMR and 13C-NMR analysis confirmed the molecule of riboflavin extract was consistent with VB2(C17H20N4O6). This is enough to show that the yellow compound in yellow mushroom is easily soluble in water, and the only compound that appears under the yellow wavelength absorption peak is riboflavin.

### 3.3. Riboflavin Synthesis Also Exists in White Mushroom

Differentially accumulated compounds were mainly enriched in two metabolic pathways, namely, Glycerophospholipid Metabolism and Purine Metabolism. The KEGG enrichment map shows that Purine Metabolism has the most differential metabolic compounds (Appendix A). The Purine Metabolism downstream product is also the riboflavin synthesis substrate GTP [38,39]. In the riboflavin biosynthetic metabolic pathway, only rib2 increased by 1.88-fold; the intermediate product 5-amino-6-ribitylamino-2,4-(1H,3H)-pyrimidinedione-5′-phosphate was not differentially expressed. Furthermore, the pentose phosphate pathway was finally regulated by rib3, rib4, and ribE to form riboflavin. Interestingly, rib3, rib4, RFK, and FLAD1 were upregulated in yellow mushrooms. RibE was only expressed in white mushrooms but not in yellow mushrooms. The specific expression of ribE in white mushrooms might be the main reason for the extraction of riboflavin from white mushrooms, although the content of extracted riboflavin was not as high as that from yellow mushrooms [40,41]. The regulation of structural genes related to riboflavin metabolism can only explain the low riboflavin content in white mushrooms and cannot clarify the yellow phenotype; hence, it is likely that the yellow phenotype might be closely related to cell transport.

### 3.4. FlMCH5 Produces the Yellow Phenotype

In the early stages, the genome assembly of yellow mushroom was completed, and the genome of yellow mushroom is 28.7 Mb, with BUSCO reaching 93.9% [42]. Evolutionary analysis showed that yellow mushroom was near *Agaricus bisporus*, and FlMCH5 could also be retrieved in the yellow mushroom genome library, but it did not exist in *Agaricus bisporus* or white mushroom in this study.

In *S. cerevisiae*, proline degradation has been reported to be closely associated with riboflavin transport [43,44], while only glycyl-L-proline was downregulated in the metabolome of yellow mushrooms, with a log_2_ fold change value reaching −1.32. Furthermore, in *S. cerevisiae*, proline degradation can activate the PUT3 transcription factor [45]. PUT3 regulates the other two proline-regulating genes, PUT1 and PUT2, and can also positively control the riboflavin transporter MCH5 [46]. In the present study, transcriptome comparison revealed that the homologous genes of PUT3 and FlMCH5 in yellow mushrooms were Unigene2188_All and CL911.Contig3_All, respectively, and that these two transcripts were only expressed in yellow mushrooms and not in white mushrooms. While PUT2 homologous genes could be detected in all transcripts, PUT1 homologous genes could not be determined, and the expression increase in PUT2 in white mushrooms was only 0.14. Western blot analyses showed that the increased riboflavin expression in the mutant strain was due to the activation of MCH5 mediated by PUT3. The gel shift test proved that PUT3 can bind to the promoter of MCH5. It must be noted that PUT3-mediated transcriptional activation requires proline as an inducer [47]. Thus, downregulation of proline in chanterelles resulted in the upregulation of the transcription factor PUT3 (Unigene2188_All), which activated the riboflavin transporter FlMCH5 (CL911.Contig3_All) that transported extracellular riboflavin into the cell, leading to the yellow phenotype (Appendix A).

### 3.5. FlMCH5 Is a Functional MFS Transcription Factor Gene Regulation the Yellow Phenotype

FlMCH5 belongs to MFS and was homologous to the riboflavin transporter MCH5 in Hypsizygus marmoreus. MFS is one of the largest families of transporters. Analysis of the amino acid structure of 66 genes revealed that the gene with the widest coverage was 7bp3.1.A, which had a structure that conformed to the condition that each domain of the MFS protein was composed of inverted 3 + 3 repeats [47]. Subcellular localization showed that FlMCH5-GFP acted more on the cell wall, further proving that the gene was related to membrane transport. Although there is no phenotypic change in transgenic tobacco, there are significant differences in riboflavin extract in leaves. The color of the leaf extract of transgenic tobacco is deeper, and the riboflavin content is higher. All of these results implied that FlMCH5 was a functional MFS transcription factor regulating riboflavin biosynthesis.

### 3.6. Creation of a New Disease-Resistant Tobacco with a High Riboflavin Content

Riboflavin in tobacco was discovered as early as the late 1950s; as the elicitor of tobacco defense response, riboflavin can enhance the resistance of tobacco to diseases such as fungi, bacteria, and viruses. At the same time, riboflavin in tobacco can be used as an antioxidant, which helps prevent the conversion of nitrate in the smoke, and can also be used as a photosensitizer to catalyze the oxidation and degradation of nicotine by antibodies. The content of riboflavin in tobacco from Guizhou, Sichuan, and other places in China determined by LC-ITMS was 5.81, 5.77, and 8.50 μg/g, respectively. In this study, the riboflavin content in tobacco overexpressing the FlMCH5 gene was as high as 26 μg/g–36.94 μg/g. Compared with wild-type or other domestic tobacco, the riboflavin content increased by 5–7 times. Subsequent cross-breeding work has a high probability of cultivating a new type of tobacco with strong disease resistance.

## 4. Materials and Methods

### 4.1. Plant Materials

The yellow mushroom (*Floccularia luteovirens*) (nwipbYM-1807) and white mushroom (*Agaricus campestris*) (nwipbYM-1808) were collected from the Qilian Mountains in Qinghai Province, China (latitude: 39°24′ N, longitude: 98°53′ E, altitude approximately 3800 m). Two kinds of mushrooms were identified by morphological identification as *Floccularia luteovirens* and *Agaricus campestris*, and the collected samples were stored at Northwest Plateau Institute of Biology, Chinese Academy of Sciences, China. White mushroom is a white mutant within 100 m of the colony circle of yellow mushroom. No permission was required for sample collection.

### 4.2. Analysis of Chemical Contents

In this study, white mushroom and yellow mushroom were selected for genetic analysis under the same environmental conditions. The white and yellow mushroom samples were frozen in liquid nitrogen immediately after collection. Before chemical and transcriptomic analysis, the white and yellow caps were stored in a refrigerator at −80 °C. The lyophilized sample was crushed at 30 Hz for 1 min using a steel ball crushing mill (Retsch MM 400, Haan, Germany). Then, 100 mg of the resultant powder was extracted with 1.0 mL of pure water overnight at 4 °C in dark. Then, the solution was centrifuged at 12,000× *g* for 10 min, and the extract was absorbed (CNWBOND Carbon-GCB SPE column, 250 mg, 3 mL; ANPEL, Shanghai, China, www.anpel.com.cn/cnw (accessed on 28 June 2019)), filtered (SCAA-104, pore size, 0.22 μm; ANPEL, Shanghai, China, http://www.anpel.com.cn/ (accessed on 28 June 2019), and subjected to liquid chromatography–mass spectrometry (LC/MS) analysis. Then, the extracts were analyzed using LC electrospray tandem MS (ESI MS/MS) system (Applied Biosystems 4500 QTRAP, http://www.appliedbiosystems.com.cn/ (accessed on 29 June 2019)) with water (0.01% acetic acid) and acetonitrile (0.01% acetic acid) as mobile phases [48]. The gradient program was as follows: 5% acetonitrile at 0 min, 20% acetonitrile at 3 min, 30% acetonitrile at 8 min, and 90% acetonitrile at 13 min; flow rate of 0.40 mL/min; temperature of 40 °C; and injection volume of 2 μL.

The raw data of LC/MS were converted into CDF format (NetCDF) using Agilent 5975c data analysis software and then preprocessed by XCMS (www.bioconductor.org (accessed on 8 August 2019)) to export the three-dimensional data matrix in CSV format [49,50]. Unsupervised principal components (PCA) were used to study the overall distribution between samples and stability of the entire analysis process. Subsequently, supervised partial least squares analysis (PLS-DA) was employed to distinguish the metabolic profiles between the groups. The model was effective if the overall difference and R2 intercept were less than 0.4 and Q2 intercept was less than 0.05; otherwise, it was defined as simulation overfitting. After determining the model, the difference in metabolites between the groups was analyzed. In the PLS-DA, the variable weight value (VIP) was higher than 1, and the variable that was less than 0.05 through *t*-test was considered to be the difference variable. To prevent the PLS-DA model from overfitting, the method of seven cycles of interactive verification and 999 tests of response ranking was used to examine the quality of the model [51,52]. R (www.r-project.org/ (accessed on 23 August 2019)) thermogram cluster analysis was used to obtain more obvious metabolite distribution [53].

### 4.3. Qualitative and Quantitative Analysis of Yellow Compound

Using a vacuum freeze dryer, the lyophilized sample was weighed, ground into a crushed residue, mixed with adequate amount of sterile water, and sonicated at 50 °C for 30 min in a water bath. Then, the supernatant yellow extract was pipetted, filtered through a 0.45 μm membrane, and subjected to HPLC under the following conditions: 0–5 min, 5–20% acetonitrile; 5–30 min, 20–25% acetonitrile; 30–35 min, 25–90% acetonitrile; column temperature, 25 °C; injection volume, 50 μL; operation time, 35 min; and spectral scanning value, 444 nm [54]. The target peak was selected for preparation and recovery. The injection volume was 1.8 mL, number of injections was 50 times, and detection wavelength was 444 nm. The prepared solution was concentrated and freeze-dried to obtain a dry powder of riboflavin extract [55]. A total of 0.05 g of riboflavin standard (Vb2: Yuanye Biology, CAS#83-88-5, HPLC ≥ 99%, experimental concentration 1%) was dissolved in 2 mL methanol for preparation liquid check. The riboflavin content in the yellow and white mushrooms was determined according to national standards (GB5009.85-2016). The concentration of Vb2 standard stock solution was 2.5 mg/g; we diluted the concentration to 0.02, 0.04, 0.06, and 0.08 mg/g; and the injection volume was 10 µL. Y = 4570.3X and R^2^ = 0.9989.

### 4.4. Transcriptome Sequencing and Analysis

The white and yellow mushroom caps were frozen at −80 °C, and the total RNA was extracted according to sequencing requirements. Mushrooms RNA was extracted by using TIANGEN polysaccharide and polyphenol extraction kit (CAT#:DP441, Beijing, China. TIANGEN Biotech Company). A total of 1 μL of extracted RNA solution was aspirated, and the RNA concentration was determined using NanoDrop. The ratio of A260/A280 was about 1.8–2.1. The integrity and high purity of the extracted RNA were determined using NanoDrop, Qubit2.0, and Agilent 2100 Bioanalyzer [56]. After qualitative analysis of the samples, the sequencing library was constructed. The selected cDNA was sequenced using Illumina high-throughput sequencer (X-10) [57]. After obtaining high-quality sequencing data, Trinity was employed for sequence assembly. The k-mer library was constructed by interrupting the reads according to the short k-mer fragment (k-mer) [58]. We used HISAT2 v2.1.0 to align the quality-controlled transcriptome reads to the yellow mushroom (*Floccularia luteovirens*) genome (https://doi.org/10.6084/m9.figshare.19354028 (accessed on 15 October 2019)).

BLAST software was utilized to compare the Unigene sequence with NR, Swiss-Prot, GO, COG, KOG, and KEGG databases; HMMER software was employed to compare the Unigene sequence with Pfam database after predicting the amino acid sequence of Unigene; and the annotation information of Unigene was obtained [59]. By selecting BLAST parameter E-value less than 1 × 10^−5^ and HMMER parameter E-value less than 1 × 10^−10^, Unigene with annotation information was finally acquired. Bowtie was employed to compare the sequenced reads with the Unigene library and combined with RSEM to evaluate the expression level. FPKM value was used to indicate the expression abundance of the corresponding Unigene. As the identification standard for the screening process, the FDR was less than 0.001, and the difference in expression between the two groups’ fold change was higher than 4 [60].

### 4.5. Functional Analysis of FlMCH5

The amino acid structure was analyzed using transmembrane transporter (https://swissmodel.expasy.org/interactive (accessed on 7 March 2020)), and the protein was identified (https://www.rcsb.org/structure (accessed on 7 March 2020)). The MEGA7 software was used for the development of tree drawings [61]. The “TAG” sequence of the FlMCH5 gene was removed and constructed in a pK7WGF-2 expression vector with the GUS gene by using gateway technology. After shaking Agrobacterium overnight, the supernatant was centrifuged to remove the supernatant, and the permeate was added. After diluting 10 times, the OD600 value was measured and controlled at about 0.4, and it was allowed to stand at room temperature for 1–3 h. We put the tobacco under a white fluorescent lamp for 1 h, injected the suspended bacteria liquid into the back of the tobacco using a syringe, and cultivated it in the dark for 2–3 days after the injection. We cut off the infected area and observed with green fluorescence after filming [62].

### 4.6. Overexpression of FlMCH5 in Tobacco

The full length of FlMCH5 was cloned by PCR; the primers for the selected genes were designed by Vector NTI10.0 (Appendix A) [63]. The sequence was cloned into the pEASY^®^-Blunt vector (TransGen Biotech, Beijing, China), which transformed into Escherichia coli. DH5α cell and the positive cloned were sequenced. FlMCH5 plasmid and PC2300s plasmid were double-digested using XbaI and BamHI. The PC2300s:FlMCH5 plasmid was transformed into GV3101. Nicotiana tabacum (Samsun) was used for transformation using the leaf disc transformation method [64]. Quantitative analysis of riboflavin in transgenic plants was performed using the same method as for yellow mushrooms.

## 5. Conclusions

This study used the combined analysis of the transcriptome and metabolome to explain the molecular mechanism of the formation of yellow mushroom, a characteristic fungus resource in Northwest China. We found that the formation of the yellow phenotype was strongly associated the accumulation of riboflavin and the regulation of the transporter FlMCH5, which transports riboflavin from outside the cell into the cell, thus creating a yellow phenotype.

This study not only confirmed that yellow mushroom has a variety of amino acids but also found that Vb2(riboflavin) laid a sloid foundation for the yellow phenotype. Our research provides a new opportunity for advertising the yellow mushroom in Northwest China. In addition to yeast, tobacco can also be used as one of the transporter function analysis carriers.

## Figures and Tables

**Figure 1 jof-08-00314-f001:**
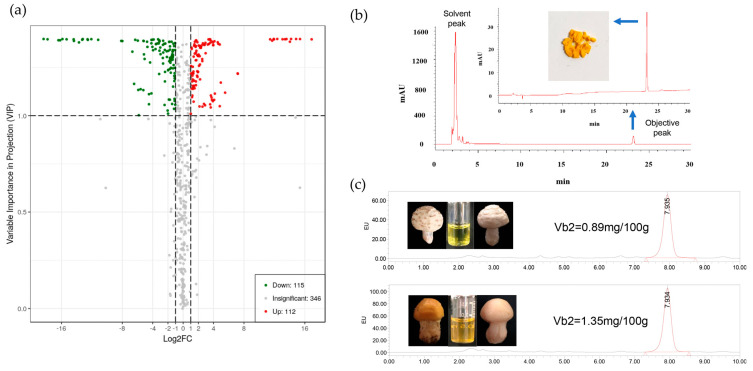
Compound analysis of yellow mushroom. (**a**) Scatter plot of differential accumulation compounds, where red dots represent compounds with high content in yellow mushroom. (**b**) Full scan chromatogram of yellow mushroom at 444 nm after ultrasonic extraction with water. The compound was prepared and freeze-dried for about 23.128 min. (**c**) The white and yellow mushrooms were ultrasonically extracted with water until presentation of the white phenotype, and then the riboflavin content was determined, respectively.

**Figure 2 jof-08-00314-f002:**
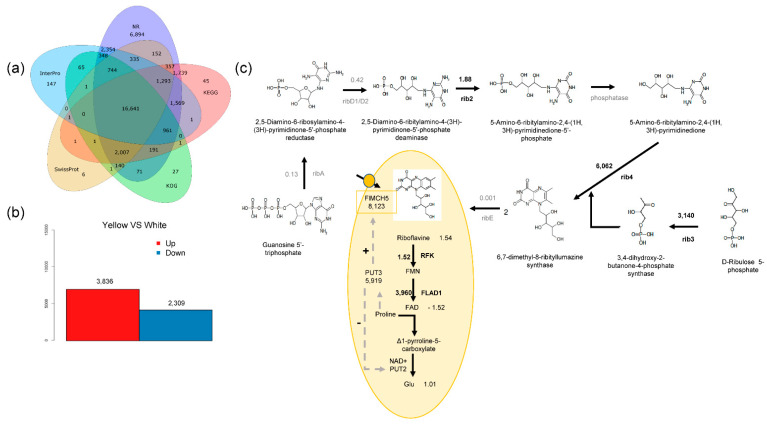
Comparative analysis of transcriptome between yellow and white mushrooms. (**a**) BLAST software was utilized to compare the Unigene sequence with NR, Swiss-Prot, GO, COG, KOG, and KEGG databases. (**b**) Differentially expressed genes between yellow and white mushrooms, where red represents the number of genes upregulated in white mushroom. (**c**) Simulation diagram of riboflavin metabolic pathway and riboflavin transport in cells. Arrows represent the direction of the composite, “+” represents that PUT3 positively regulates FlMCH5, and “–” represents that PUT3 negatively regulates PUT2.

**Figure 3 jof-08-00314-f003:**
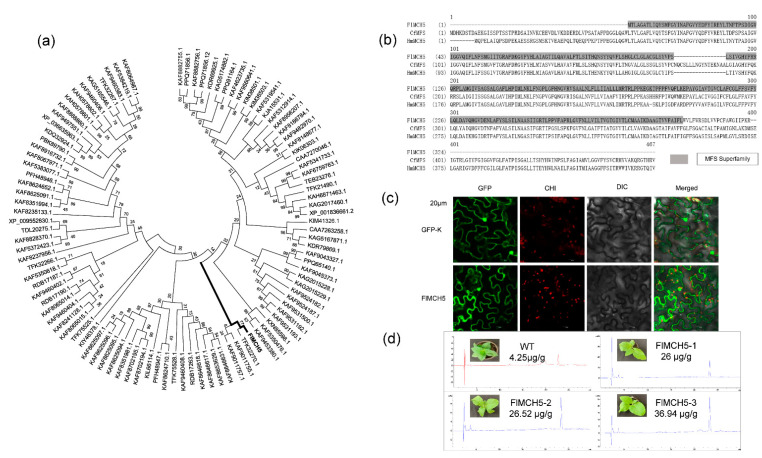
Functional analysis of FlMCH5. (**a**) Phylogenetic tree analysis of FlMCH5. (**b**) FlMCH5 structural–functional domain comparison, where gray represents MFS superfamily. (**c**) The expression of GUS protein negative control and FlMCH5-GUS in tobacco cells; the cell size in the picture is 20 μm. (**d**) Riboflavin was extracted from transgenic tobacco leaves; the content of riboflavin was determined to be 3 g in leaves.

**Table 1 jof-08-00314-t001:** Summary of differently accumulation chemical compounds in yellow and white mushrooms.

Class	Total	Down (in Yellow)	Up (in Yellow)	Number Different	Ratio (Yellow/White)
Amino acid and derivatives	98	22	23	45	0.466
Phenylpropanoids	35	13	7	20	0.278
Flavone	35	1	8	9	8.758
Flavonol	16	2	1	3	3.691
Flavonoid	16	1	1	2	0.339
Flavanone	11	2	1	3	1.392
Isoflavone	2	1	1	2	2.149
Alcohols	15	5	0	5	0.005
Polyphenol	12	1	1	2	0.046
Phenolamides	12	2	4	6	1.969
Nucleotide and derivates	59	20	13	33	1.167
Others	27	6	6	12	0.387
Alkaloids	28	8	0	8	0.103
Carbohydrates	22	2	8	10	2.446
Terpene	11	0	3	3	2.425
Vitamins and derivatives	17	1	6	7	3.362
Indole derivatives	7	3	0	3	0.315
Organic acids and derivatives	81	12	11	23	1.132
Lipids	61	12	18	30	2.274
Anthocyanins	2	1	0	1	0
Quinones	1	0	0	0	/
Sterides	5	0	0	0	/
sum	573	115	112	227	

## Data Availability

The transcriptomic data were successfully uploaded to NCBI, PRJNA647230: RNA-seq data of *Agaricus campestris*, PRJNA647227: RNA-seq data of *Floccularia luteovirens*. All data generated or analyzed during this study are included within the article and its additional files.

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
