# Peer review of "Chemical Constituents and Molecular Mechanism of the Yellow Phenotype of Yellow Mushroom (Floccularia luteovirens)"

_jof, 2022, doi:10.3390/jof8030314_

Round 1

Reviewer 1 Report

This is a very interesting study. This manuscript needs some additions and corrections.

  1. How is the genetic relationship between yellow and white mushrooms? Are the yellow mushroom and white mushroom different species? (see lines 334-335).
  2. The authors analyzed white mushrooms for color analysis. Why didn't you analyze mushrooms of different colors?
  3. It is thought that the results of qPCR to validate the quality of transcriptome sequencing are necessary in this manuscript. 

Author Response

Dear reviewers,

        Thanks very much for your constructive comments, which have greatly helped the revision of our work. We have considered your comments seriously, and have taken relevant actions to improve the manuscript. The efforts we have made are explained below your comments, and they have also been incorporated into the revised manuscript in an appropriate manner. We hope that this revision would be satisfactory.

Best regards,

Zong Yuan (on behalf of all co-authors)

  1. How is the genetic relationship between yellow and white mushrooms? Are the yellow mushroom and white mushroom different species? (see lines 334-335).

Responses: Thanks for this suggestion. Both are Agaric, Mushroom family, but different genera. So far, yellow mushrooms cannot be cultivated artificially. This study can only select similar species under the same environmental conditions as the control group.

  1. The authors analyzed white mushrooms for color analysis. Why didn't you analyze mushrooms of different colors?

Responses: Thanks for this suggestion. The white mushrooms and yellow mushrooms were selected for genetic analysis under the same background conditions. The follow-up work will also add analysis and research for other colors.

  1. It is thought that the results of qPCR to validate the quality of transcriptome sequencing are necessary in this manuscript. 

Responses: Thanks for this suggestion. Previous genomic analysis of Floccularia luteovirens had confirmed that the expression level of FlMCH5 is consistent with the three repeats transcriptome results, so no qPCR was added.

Reviewer 2 Report

This study uses the combined analysis of transcriptome and metabolome to explain the molecular mechanism of the formation of the yellow phenotype of yellow mushroom. Authors demonstrated that riboflavin is the main coloring compound of yellow mushrooms, and FlMCH5 is the key transport regulatory gene that produces the yellow phenotype.

Although the manuscript is carefully written and the presented experiments seem to be well conducted, however, there are few questions to the authors.

  1. What methods (morphological, genetic) were used to the species identification of collected samples? Was the taxonomical verification of the samples was done using DNA sequence analysis?

Line 343 – 344. "White mushroom is a white mutant within 100 meters of the colony circle of yellow mushroom". What does it mean? Previously (line 339 – 340), you wrote that the yellow mushroom is Floccularia luteovirens and white mushroom is Agaricus campestris. These are two different species belonging to different genera. Do you mean, Floccularia luteovirens is white mutant of Agaricus campestris or this is inaccurate translation?

What do you mean by a "variety"?

Line 101, "between the two varieties of mushrooms"  Varieties or species of mushrooms?

Line 115. Varieties or species?

  1. Please check the matching of file numbers with numbers of tables.

Table S3 (separate file Table S3.docx) is identical to Table 1.

File TableS7.xls contains Table S8

File TableS8.xls contains Table S9

File TableS9.xls contains Table S10

I cannot find TableS7

  1. Tables require some explanations in captures:

Table 1. I recommend adding a capture to the table with explanations what is time(Y/W), down, up.  I consider, "time(Y/W)" is better to call "Ratio(Yellow/White)" or "Fold Change(Yellow/White)"

Table S2. I cannot find table capture. W-1, W-2, W-3 are samples of white mushrooms and Y-1 – 3 are samples of yellow mushrooms, I consider? Columns D – Y: what are units of these data?

  1. Line 65 – 74. It is unclear from the Introduction if MCH5p is the member of the MFS class.

  1. Line 93 – 95. "The contents of 112 compounds were higher in yellow mushroom" – this statement is complies with the table S2. Please clarify the next sentence: "The contents of 11 compounds in yellow mushrooms were higher than those in white mushrooms"

11 or 112 compounds were higher in yellow mushroom?

  1. Abbreviations should be defined the first time they appear:

Line 59 – 64. Abbreviations DARPP and ArP.

Line 149 Abbreviation DEGs should be defined: differentially expressed genes (DEGs)

  1. Figure 2C. The resolution of this interesting figure in PDF version is not enough to read the text and chemical formulas, some bonds in formulas are not visible. Figure 3A and B – the same problems with resolution in PDF.

  1. Please make some clarifications by the methodology:

Line 358 – 359. Please provide the name and the manufacturer of the mass-spectrometer.

Line 398. Please specify the method of RNA extracion.

  1. Please go through the text thoroughly and correct syntax errors:

Line 65 – 66. Rephrase "large class…have 12 …domains". Not class but proteins included in the class have 12 domains.

Line 78. "…the yellow main effect compound…" It seems that the phrase is incorrect.

Line 101. "differentially expressed flavonol compounds"  - Differentially accumulated, not expressed.

Line 118. After water extraction of…

Line 188. I would say "In other species" or "fungal species"

Line 245. Chanterelles? Fungi of the order Cantharellales? It's not a mistake?

Lines 347 – 349. I did not understand. Rephrase please.

Line 354. 1mL of pure water or aqueous solution of any dissolved substance?

Line 191. Fungal names should be italics.

Author Response

Dear reviewers,

        Thanks very much for your constructive comments, which have greatly helped the revision of our work. We have considered your comments seriously, and have taken relevant actions to improve the manuscript. The efforts we have made are explained below your comments, and they have also been incorporated into the revised manuscript in an appropriate manner. We hope that this revision would be satisfactory.

Best regards,

Zong Yuan (on behalf of all co-authors)

  1. What methods (morphological, genetic) were used to the species identification of collected samples? Was the taxonomical verification of the samples was done using DNA sequence analysis?

Line 343 – 344. "White mushroom is a white mutant within 100 meters of the colony circle of yellow mushroom". What does it mean? Previously (line 339 – 340), you wrote that the yellow mushroom is Floccularia luteovirens and white mushroom is Agaricus campestris. These are two different species belonging to different genera. Do you mean, Floccularia luteovirens is white mutant of Agaricus campestris or this is inaccurate translation?

What do you mean by a "variety"?

Line 101, "between the two varieties of mushrooms"  Varieties or species of mushrooms?

Line 115. Varieties or species?

 Responses: Thanks for this suggestion.

Firstly: “Morphological identification” had been added in “Plant Materials”.

Secondly: The white mushrooms and yellow mushrooms were selected for genetic analysis under the same background conditions.

Thirdly: They are two different species, but both are two species with different colors that grow in the same environment, suitable for difference comparison analysis. So far, yellow mushrooms cannot be cultivated artificially.

  1. Please check the matching of file numbers with numbers of tables.

Table S3 (separate file Table S3.docx) is identical to Table 1.

File TableS7.xls contains Table S8

File TableS8.xls contains Table S9

File TableS9.xls contains Table S10

I cannot find TableS7

  Responses: Thanks for this suggestion. This error and Tables order had been revised.

  1. Tables require some explanations in captures:

Table 1. I recommend adding a capture to the table with explanations what is time(Y/W), down, up.  I consider, "time(Y/W)" is better to call "Ratio(Yellow/White)" or "Fold Change(Yellow/White)"

Table S2. I cannot find table capture. W-1, W-2, W-3 are samples of white mushrooms and Y-1 – 3 are samples of yellow mushrooms, I consider? Columns D – Y: what are units of these data?

 Responses: Thanks for this suggestion. Table1 and Table S2 had been revised, and the values of yellow and white respectively represent the peak area of each compound in the HPLC-MS/MS instrument.

  1. Line 65 – 74. It is unclear from the Introduction if MCH5p is the member of the MFS class.

  Responses: Thanks for this suggestion. More introduction had been added for explaining the relationship between MCH5 and MFS.

  1. Line 93 – 95. "The contents of 112 compounds were higher in yellow mushroom" – this statement is complies with the table S2. Please clarify the next sentence: "The contents of 11 compounds in yellow mushrooms were higher than those in white mushrooms"

11 or 112 compounds were higher in yellow mushroom?

  Responses: Thanks for this suggestion. It had been revised as “11 classes of compounds”.

  1. Abbreviations should be defined the first time they appear:

Line 59 – 64. Abbreviations DARPP and ArP.

Line 149 Abbreviation DEGs should be defined: differentially expressed genes (DEGs)

  Responses: Thanks for this suggestion. These Abbreviations had been added.

  1. Figure 2C. The resolution of this interesting figure in PDF version is not enough to read the text and chemical formulas, some bonds in formulas are not visible. Figure 3A and B – the same problems with resolution in PDF.

   Responses: Thanks for this suggestion. These pictures had been revised.

  1. Please make some clarifications by the methodology:

Line 358 – 359. Please provide the name and the manufacturer of the mass-spectrometer.

Line 398. Please specify the method of RNA extracion.

   Responses: Thanks for this suggestion. More introductions had been added.

  1. Please go through the text thoroughly and correct syntax errors:

Line 65 – 66. Rephrase "large class…have 12 …domains". Not class but proteins included in the class have 12 domains.

Line 78. "…the yellow main effect compound…" It seems that the phrase is incorrect.

Line 101. "differentially expressed flavonol compounds"  - Differentially accumulated, not expressed.

Line 118. After water extraction of…

Line 188. I would say "In other species" or "fungal species"

Line 245. Chanterelles? Fungi of the order Cantharellales? It's not a mistake?

Lines 347 – 349. I did not understand. Rephrase please.

Line 354. 1mL of pure water or aqueous solution of any dissolved substance?

Line 191. Fungal names should be italics.

   Responses: Thanks for this suggestion. All these errors had been revised.

Reviewer 3 Report

The authors investigated the molecular mechanism determing the yellow phenotype of an important mushroom using the metabolom and transcriptome analysis. However, the English writing, the method and results interpretation need hugely improvements.  

My major concerns are listed below: 

Why the well-sequenced genome “Floccularia luteovirens” was not used in this study to map the transcriptome data? In addition, the well sequenced and annotated white-mushroom “Agaricus bisporus” could be a better control sample than the “Agaricus campestris” described in this manuscript. The transcriptome assembly based on Trinity analysis caused some bias or mistakes in the results (e.g. Fig.S4 has weird KEGG terms: Parkinson disearse and taste transduction...).

The HPLC data, GFP and overexpression experiments partially support that the riboflavin and related transporter contribute to the yellow color of the mushroom. However, the author should draw conclusion very carefully, given that some other compounds are present in the metabolome and the direct knock-down/deleting the corresponding gene in studied or closely related mushroom hasn’t been performed yet. 

The English writing is quite poor. For examples, Line 16, mushroom was considered as value ‘crop’; Line 20, there is a gramma mistake as the verb is missing. Line 24, Log2fold is not a proper word. Line 27, the description of three concentrations of riboflavin is a bit confusing.  

Author Response

Dear reviewers,

        Thanks very much for your constructive comments, which have greatly helped the revision of our work. We have considered your comments seriously, and have taken relevant actions to improve the manuscript. The efforts we have made are explained below your comments, and they have also been incorporated into the revised manuscript in an appropriate manner. We hope that this revision would be satisfactory.

Best regards,

Zong Yuan (on behalf of all co-authors)

The authors investigated the molecular mechanism determing the yellow phenotype of an important mushroom using the metabolom and transcriptome analysis. However, the English writing, the method and results interpretation need hugely improvements.  

My major concerns are listed below: 

Why the well-sequenced genome “Floccularia luteovirens” was not used in this study to map the transcriptome data? In addition, the well sequenced and annotated white-mushroom “Agaricus bisporus” could be a better control sample than the “Agaricus campestris” described in this manuscript. The transcriptome assembly based on Trinity analysis caused some bias or mistakes in the results (e.g. Fig.S4 has weird KEGG terms: Parkinson disearse and taste transduction...).

Responses: Thanks for this suggestion.

1.Analysis methods and genomic information had been added in “Materials and Methods”.

  1. The white mushrooms and yellow mushrooms were selected for genetic analysis under the same background conditions.
  2. Because the Fungi library is relatively small, we annotated all genes into the entire KEGG library, but mushrooms only accounted for 50% of the genes annotated in KEGG, so some genes were annotated in pathways related to human diseases (Parkinson disearse)

The HPLC data, GFP and overexpression experiments partially support that the riboflavin and related transporter contribute to the yellow color of the mushroom. However, the author should draw conclusion very carefully, given that some other compounds are present in the metabolome and the direct knock-down/deleting the corresponding gene in studied or closely related mushroom hasn’t been performed yet. 

Responses: Thanks for this suggestion.

We used the metabolome to determine the presence of sakuranetin, vanillic acid, skimmine, curdione and other highly medicinal compounds in yellow mushrooms, and we will conduct more research on the accumulation of gene regulatory compounds in the next work.

The English writing is quite poor. For examples, Line 16, mushroom was considered as value ‘crop’; Line 20, there is a gramma mistake as the verb is missing. Line 24, Log2fold is not a proper word. Line 27, the description of three concentrations of riboflavin is a bit confusing.  

Responses: Thanks for this suggestion.

These errors had been revised and the English writing had been complete modification under “editing service” by MDPI.

Round 2

Reviewer 3 Report

I saw a clear improvement in the updated version. I have only a few comments:

The author could add some introduction about the common known molecular mechanism of colors in mushroom or other organisms. Although Vb2(riboflavin) is a well-studied yellow component, it's not considered as the main factor determining the color for most flower and plant. I still suggest that the conclusion should be more conservative. We can't fully exclude other factors may contribute the yellow phenotype due to the limit of current metabomoe technology and lacking of conclusive experimental validation in this study.

Line 464, is mushroom a characteristic plant resource ?  

Author Response

Dear reviewers,

        Thanks very much for your constructive comments, which have greatly helped the revision of our work. We have considered your comments seriously, and have taken relevant actions to improve the manuscript. The efforts we have made are explained below your comments, and they have also been incorporated into the revised manuscript in an appropriate manner. We hope that this revision would be satisfactory.

Best regards,

Zong Yuan (on behalf of all co-authors)

I saw a clear improvement in the updated version. I have only a few comments:

The author could add some introduction about the common known molecular mechanism of colors in mushroom or other organisms. Although Vb2(riboflavin) is a well-studied yellow component, it's not considered as the main factor determining the color for most flower and plant. I still suggest that the conclusion should be more conservative. We can't fully exclude other factors may contribute the yellow phenotype due to the limit of current metabomoe technology and lacking of conclusive experimental validation in this study.

Line 464, is mushroom a characteristic plant resource ?  

Responses: Thanks for this suggestion.

  1. These introduction (molecular mechanism of colors in organism), discussion(Vb2 was the major compound) and conclusion had been revised in manuscript.
  2. “Plant” had been revised to “fungus”.